# C(sp³)−H sulfinylation of light hydrocarbons with sulfur dioxide via hydrogen atom transfer photocatalysis in flow

Dmitrii Nagornîi [1,2], Fabian Raymenants [1,2], Nikolaos Kaplaneris[1] & Timothy Noël [1] ✉

Sulfur-containing scaffolds originating from small alkyl fragments play a crucial role in various pharmaceuticals, agrochemicals, and materials. Nonetheless, their synthesis using conventional methods presents significant challenges. In this study, we introduce a practical and efficient approach that harnesses hydrogen atom transfer photocatalysis to activate volatile alkanes, such as isobutane, butane, propane, ethane, and methane. Subsequently, these nucleophilic radicals react with $SO_2$ to yield the corresponding sulfinates. These sulfinates then serve as versatile building blocks for the synthesis of diverse sulfur-containing organic compounds, including sulfones, sulfonamides, and sulfonate esters. Our use of flow technology offers a robust, safe and scalable platform for effectively activating these challenging gaseous alkanes, facilitating their transformation into valuable sulfinates.

Gases, being lightweight and arguably the most atom-efficient choice of reagents for various transformations, present a unique set of challenges when it comes to their practical use in the laboratory environment[1]. The inherent difficulties in handling gases using conventional batch equipment, coupled with the physical separation of gases and liquid reaction mixtures due to gravity, often discourage their utilization. When gases serve as reagents, they need to diffuse within the reaction mixture before they can react effectively. To overcome this limitation, researchers frequently turn to Henry's law, which allows for the enhancement of gas solubility by increasing the pressure within the vessel[2]. However, this approach necessitates specialized equipment, such as Parr bombs, and additional safety precautions. As a consequence, setting up multiple reactions concurrently becomes impractical, resulting in a slow pace of reaction discovery and optimization. To circumvent these challenges, chemists often resort to the development of engineered reagents that are crystalline, easy to handle and store[3]. However, such reagents tend to be more expensive and less atom-efficient than their gaseous counterparts. For instance, delivering 1 mol of $SO_2$ from the convenient reagent DABSO[4–6] [i.e., DABCO-bis(sulfur dioxide)] costs approximately three orders of magnitude more than using the gas itself.

Besides practical considerations, some gases are also challenging to activate under conditions suitable for synthetic organic chemistry due to their inert nature. Notably, achieving selective functionalization of C(sp³)−H bonds in saturated volatile hydrocarbons like methane, ethane, propane, and butane represents a significant goal in modern C−H activation chemistry[7,8]. Despite recent advancements, efficiently functionalizing gaseous alkanes remains a formidable task. This challenge is exacerbated by their unfavorable thermodynamics, with bond dissociation energies (BDE) reaching up to 105 kcal/mol[9], and compounded by issues related to chemoselectivity, solvent compatibility, and solubility (Fig. 1A)[1]. Consequently, these reactions often demand harsh conditions, characterized by high temperatures and pressures, which may not align with the delicate nature of most organic molecules.

Recently, our group and other researchers have demonstrated that hydrogen atom transfer (HAT) photocatalysis offers a promising avenue for achieving mild activation conditions in the functionalization of volatile alkanes[10–12]. Although this reaction has historically posed practical challenges when conducted in batch conditions, the emergence of flow technology has introduced a convenient reactor platform[13]. This reactor design enhances the overall irradiation profile

[1]Flow Chemistry Group, Van't Hoff Institute for Molecular Sciences (HIMS), University of Amsterdam, Amsterdam, The Netherlands. [2]These authors contributed equally: Dmitrii Nagornîi, Fabian Raymenants. ✉e-mail: T.Noel@uva.nl

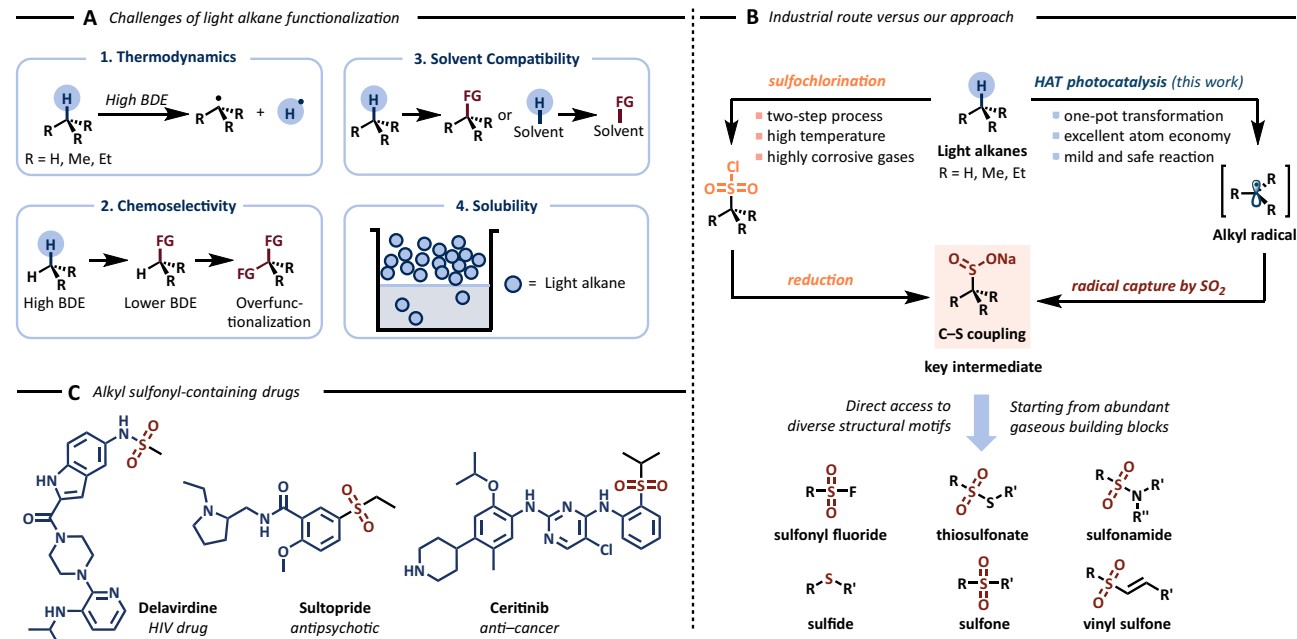

**Fig. 1 | C(sp³)−H sulfinylation of light alkanes as a strategy for the direct valorization of abundant gaseous feedstocks. A** Challenges of light alkane functionalization. **B** Industrial routes towards alkyl sulfinates in comparison with our approach **C** Examples of alkyl sulfonyl-containing drugs. BDE: bond dissociation energy, FG: functional group.

## Table 1 | Optimization of the photocatalytic sulfinylation of propane with SO₂ in flow

| Entry[a] | Parameter | Deviation from the standard conditions | Yield [%][b] |
|---|---|---|---|
| 1 | Residence time [h] | 0.5/1/2/4 | 41/48/46/25 |
| 2 | Catalyst loading [mol%] | 0.5/1/2/3 | 18/48/44/34 |
| 3 | Gas equivalents | 2.5/5/10 | 19/48/45 |
| 4 | DT source | TBADT/NaDT | 48/59 |
| 5 | Concentration SO₂ [mol/L] | 0.1/0.2 | 43/59 |
| 6 | BnBr as limiting reagent | BnBr 1 eq.[c] | 95 |

[a]Reaction conditions: SO₂ (0.4 mmol), propane (2.0 mmol), DT catalyst (1 mol%) in CH₃CN/H₂O (4:1, 2 mL, 0.2 M) irradiated in photoreactor (365 nm, 144 W optical output power) for 1 h. The outflow is collected in a flask containing NaHCO₃ (1.0 mmol) and benzyl bromide (0.6 mmol, 1.5 equiv.) and is stirred overnight at room temperature.
[b]Yield determined by ¹H-NMR spectroscopy using trichloroethylene as external standard.
[c]Optimized conditions: SO₂ (0.6 mmol), propane (3.0 mmol), DT catalyst (1 mol%) in CH₃CN/H₂O (4:1, 3 mL, 0.2 M) irradiated in photoreactor (365 nm, 144 W optical output power) for 1 h. The outflow is collected in a flask containing NaHCO₃ (1.2 mmol) and benzyl bromide (0.2 mmol, 1 equiv.) and is stirred overnight at room temperature.

within the reaction mixture and increases gas-liquid mass transfer rates[14]. As a result, this approach leads to substantial reductions in reaction times, increased selectivity[15], and enhanced practicality for exploring a wider scope of reactions[16]. Notwithstanding this recent progress, there are still significant limitations in the scope of available reactions suitable for light alkane activation.

Sulfones, sulfonamides, and sulfonate esters represent ubiquitous functional groups with applications spanning various fields, including pharmaceuticals[17,18], agrochemicals[19,20], and material science (Fig. 1C)[21]. Commonly, sulfinate salts serve as versatile precursors for constructing sulfur-containing compounds due to their reactivity and stability[22,23]. Industrial methods currently employed for synthesizing sodium sulfinates rely on a two-step process involving the reduction of corresponding sulfonyl chlorides. These chlorides, in turn, are derived

from the respective alkanes, sulfur dioxide (SO₂), and chlorine gas (Fig. 1B)[24]. This conventional approach necessitates the handling of highly hazardous and corrosive gases (SO₂ and Cl₂) at elevated temperatures and pressures. In light of the increasing use of sulfur dioxide and its surrogates in recent methodologies[25,26], our objective was to develop a method that harnesses the advantages of both flow chemistry and photocatalysis in conjunction with gaseous alkanes.

Despite the common use of SO₂ surrogates in recent photochemical methodologies (DABSO, sodium metabisulfite and sodium bisulfite), their low solubility in common organic solvents hampers the development of flow methodologies[27–30]. Although a limited amount of work has focused on the photochemical C(sp³)−H functionalization of alkanes for the generation of sulfur containing compounds[31,32], methods utilizing gaseous alkanes or flow chemistry have not been

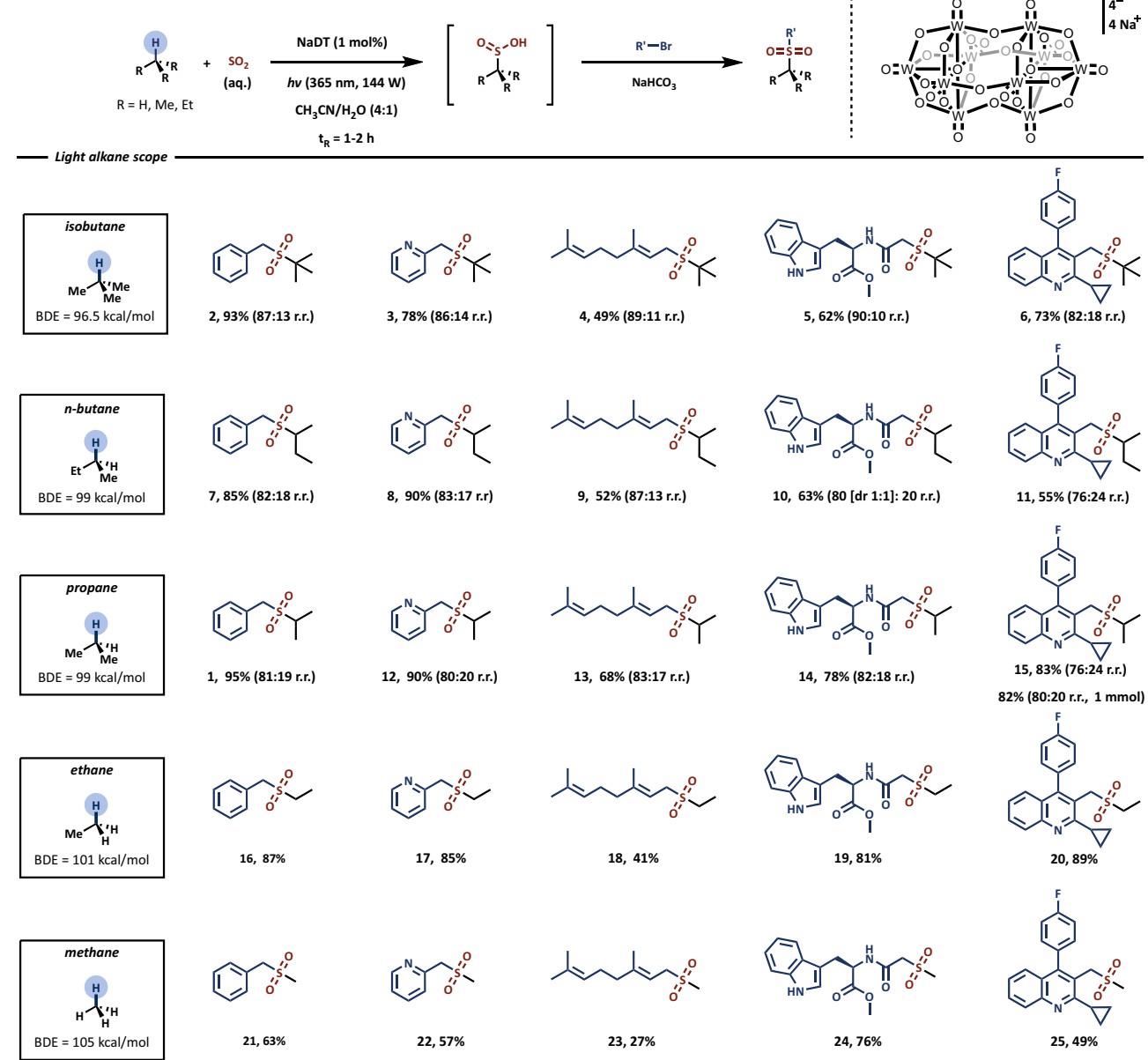

**Fig. 2 | Reaction scope for the photocatalytic sulfinylation of light alkanes with sulfur dioxide and trapping with electrophiles.** Photocatalytic reactions are performed in flow at 0.2 M concentration (SO₂, 0.6 mmol scale), with 5 equiv. of light alkane and 1 mol% of NaDT photocatalyst, under irradiation of UV-A light (365 nm, 144 W optical output power). Residence time of 1 h for isobutane, n-butane and propane, 2 h for ethane and methane. Pressure of 34 bar for isobutane, n-butane and propane, 52 bar for ethane and methane. The outflow is collected in a flask containing NaHCO₃ and alkyl bromide (0.2 mmol) and is stirred for 1 h at 60 °C. For reactions with methane, the photocatalytic step is performed at 4 mmol scale (SO₂), 6 mmol scale for entry **23** and **24**. Diastereomeric ratio of the major isomer of compound **10** is 1:1. See Supplementary Information for further experimental details.

previously reported. Alkyl sulfinates derived from C1-4 alkanes are commonly found in drug structures (Fig. 1C); however, their direct synthesis from light alkanes has proven to be an elusive task thus far. As presented herein, we have developed a general flow-based platform for generating synthetically valuable alkyl sulfinates directly from readily available and cost-effective gases (Fig. 1B). Most notably, we demonstrated their synthetic utility in the late-stage functionalization of medicinally relevant organic scaffolds.

## Results and discussion

We initiated our investigation into the proposed C(sp³)–H sulfinylation of gaseous alkanes by blending a mixture of propane and an aqueous solution of sulfur dioxide in the presence of tetrabutylammonium decatungstate (TBADT) in an acetonitrile:H₂O solution (4:1, 0.2 M).

This reaction mixture was then passed through a transparent continuous-flow microreactor (FEP tubing, ID = 0.5 mm, 1.5 mL volume) exposed to six high-intensity UV-A light sources (Chip-on-Board LED, λ = 365 nm, 144 W optical power)[33]. Subsequently, the reaction stream was combined with benzyl bromide and sodium bicarbonate using a fed-batch approach. To facilitate complete liquefaction of the gas and enhance the efficiency of C(sp³)–H bond activation through decatungstate-catalysed HAT, a 34 bar back-pressure regulator (BPR) was positioned at the reactor outlet[1]. After a careful exploration of various reaction conditions, we successfully obtained the desired product in a 48% yield, with just a one-hour residence time, in the presence of 1 mol% of TBADT (Table 1, Entries 1-2). Prolonged exposure to irradiation did not yield further improvements in the reaction yield likely due to degradation (Table 1, Entry 1). Increasing

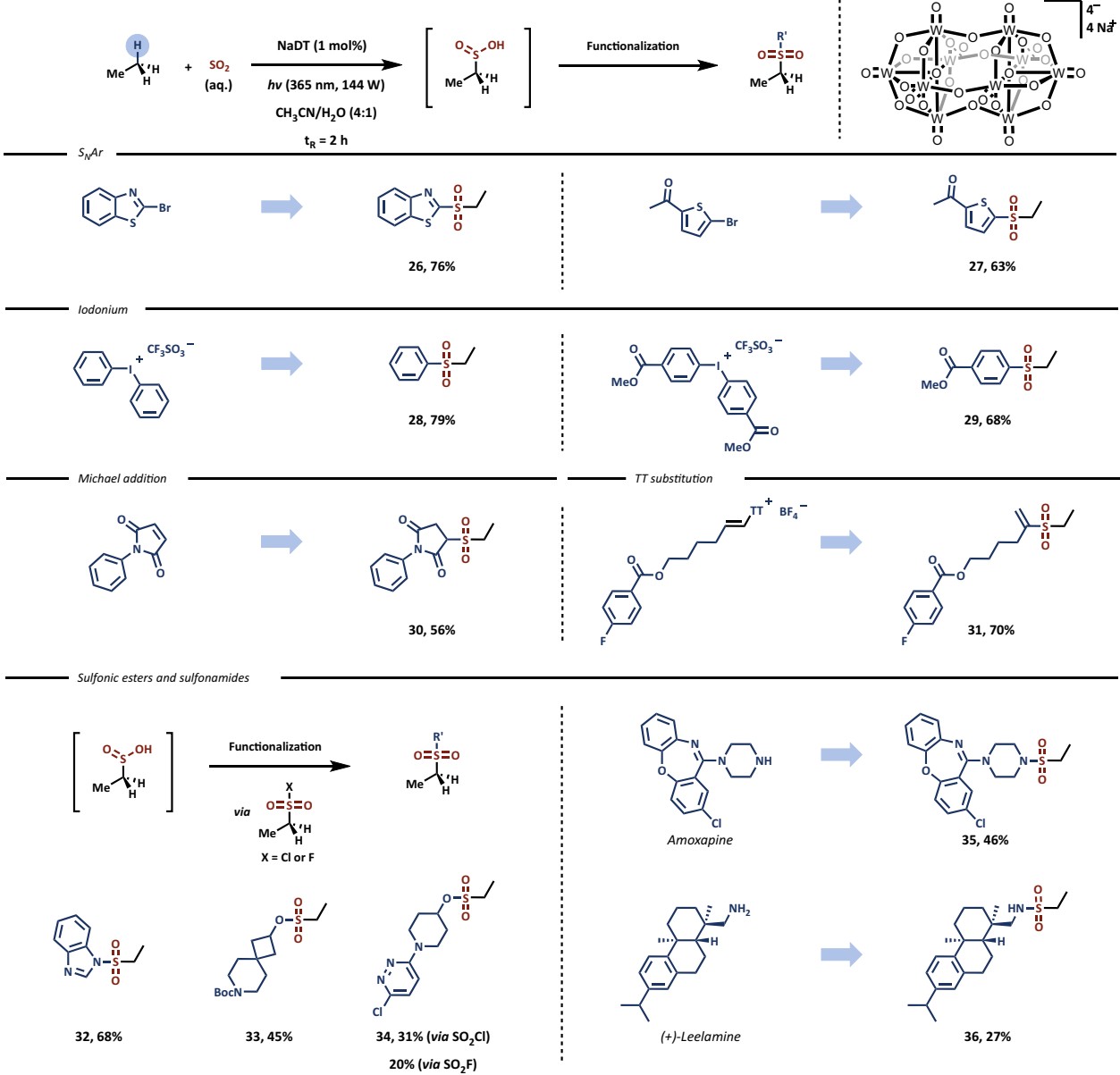

**Fig. 3 | Functionalization of ethyl sulfinic acid into diverse ethyl sulfonyl compounds.** See SI for further experimental details. TT = thianthrene.

the amount of propane gas beyond 5 equivalents did not lead to a significant improvement in yield (Table 1, Entry 3). Remarkably, the choice of cation for the decatungstate photocatalyst was found to impact the reaction yield, with sodium decatungstate (NaDT) yielding a 59% yield, likely due to its higher solubility (Table 1, Entry 4)[34]. Higher concentrations of SO₂ led to enhanced yields by improving reaction rates, simultaneously increasing the overall throughput and thus the scalability of the process (Table 1, Entry 5). Given the low cost of the gaseous components, we adjusted the stoichiometry by adding three equivalents of SO₂, resulting in a high yield for the transformation, with benzyl bromide serving as the limiting reagent (Table 1, Entry 6).

Having established optimized conditions, we embarked on exploring the sulfinylation of the light alkane homologous series (C1–C4) as depicted in Fig. 2. Within this study, we selected various electrophilic traps for the intermediate sulfinic acid, including functionally diverse alkyl bromides such as benzyl, allyl bromide, α-bromo amide and alkyl iodide (see Supplementary Information, Section 7.7). This series encompasses compounds like benzyl bromide, bromomethyl pyridine, geranyl bromide, as well as structurally intriguing

substrates such as tryptophan and quinoline derivatives, each bearing a diverse array of functional groups.

By employing isobutane as the hydrogen donor in the sulfinylation reaction, we achieved efficient installation of the *tert*-butylsulfonyl group on the series of electrophiles, yielding good to excellent results ranging from 49% to 93% isolated yield for compounds **2**–**6**. When isobutane is used, tertiary radicals are the preferred outcome over primary radicals, primarily due to the lower bond dissociation energy (BDE) of the C–H bond and the increased stability of the resulting radical. This preference results in an average regioisomeric ratio of 87:13[10].

Likewise, when butane was utilized, we achieved good to excellent yields for the same set of electrophiles (ranging from 52% to 90%). However, the regioisomeric ratio in this case was slightly lower (82:18) due to the narrower difference between the BDE values of the secondary and primary C–H bonds. Similarly, propane underwent successful functionalization, resulting in the corresponding sulfones **1, 12**–**15** in good to excellent isolated yields (ranging from 68% to 95%). Due to its comparable BDE values to butane, we observed a similar

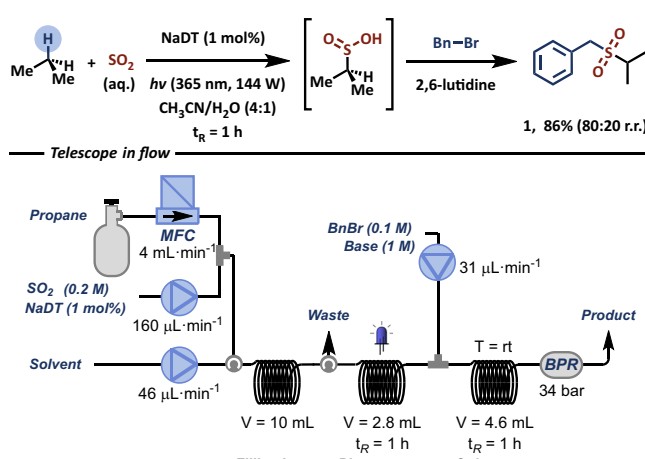

**Fig. 4 | Telescoped reaction sequence in flow of the photocatalytic sulfinylation of propane with SO₂ and subsequent trapping with benzyl bromide.** See Supplementary Information for further experimental details.

trend in regioselectivity (averaging 80:20) during the activation of propane[10].

Due to the stronger primary C(sp³)−H bonds in ethane, the functionalization process necessitated an extended residence time of 2 h instead of the previous 1 h. Additionally, to ensure complete liquefaction of the gas and minimize potential gas-to-liquid mass transfer effects, a back-pressure regulator (BPR) set at 52 bar was employed. These optimized conditions yielded synthetically useful to excellent isolated yields, ranging from 41% to 89%.

Methane, characterized by its high bond dissociation energy (BDE) of C−H bonds (BDE = 105 kcal/mol), presents a substantial challenge for functionalization. Despite requiring the most demanding activation conditions, methane stands out as one of the most abundant feedstock gases, making it a highly sought-after C1 building block[35], if it were more facile to activate at more benign reaction conditions. To tackle the functionalization of methane, a 2-h reaction time was employed, along with the use of deuterated acetonitrile to prevent the functionalization of the reaction solvent, a method aligned with our prior work[11]. These optimized reaction conditions successfully yielded synthetically valuable building blocks **21** to **25** in satisfactory to good yields, ranging from 27% to 76%.

Furthermore, we harnessed the ease of scaling up flow reactions and synthesized quinoline derivative **15** on a 1 mmol scale, maintaining both yield and regioselectivity (82%, 80:20 r.r.). This result underscores the method's ability to be seamlessly scaled up to produce meaningful quantities, which is particularly valuable for researchers working in medicinal or crop-protection chemistry.

Notably, by utilizing a single set of reaction conditions, we achieved the synthesis of structural analogues of various sulfone compounds, each modified with distinct short alkyl fragments, accomplished solely by altering the choice of the light alkane gas.

To underscore the extensive applicability of this method in the synthesis of diverse organosulfur compounds, various post-functionalization procedures were applied to ethane sulfinate (Fig. 3). Our mild protocol facilitated straightforward follow-up functionalization through an operationally-simple fed-batch process. One-pot protocols enabled the synthesis of heteroaryl sulfones via aromatic nucleophilic substitution reactions (SN_Ar), yielding benzothiazole derivative **26** and thiophene derivative **27** in 76% and 63% yields, respectively. Additionally, the use of diaryliodonium salts led to the synthesis of aryl sulfones **28** and **29** in 79% and 68%, respectively. The Michael addition reaction of ethane sulfinate intermediate with *N*-phenyl succinimide resulted in the formation of imide **30** in 56% yield. A *cine*-sulfonylation reaction was applied for the synthesis of highly

electrophilic alkenyl sulfone **31** with a yield of 70%[36]. A two-step procedure was utilized for the synthesis of sulfonamides and sulfonate esters, involving a sulfonyl halide intermediate, leading to medicinally relevant compounds **32**–**36**, including derivatizations of Amoxapine and (+)-Leelamine. Primary, secondary, and aromatic amines, along with two alcohols were sulfonylated, providing the desired products in synthetically useful yields ranging from 27% to 68%.

Lastly, we tested the unique capability of continuous-flow technology to combine different reaction steps in a telescoped reaction sequence (Fig. 4)[37]. The photocatalytic sulfinylation of propane was initiated in the first step under high-intensity light irradiation, with a residence time of 1 h. Subsequently, the process stream exiting from the photoreactor was merged with a stream containing benzyl bromide, and introduced into a second flow reactor for nucleophilic substitution. A back-pressure regulator (BPR) was positioned at the outlet of the second reactor to maintain stable flow conditions across the entire reactor network. We opted for the soluble organic base, 2,6-lutidine, instead of NaHCO₃ to prevent clogging[38]. Consequently, the corresponding sulfone product **1** was isolated in good yield after both steps (86%, 80:20 r.r), demonstrating similar efficiency to our fed-batch approach.

In conclusion, we have established a practical protocol for the photocatalytic conversion of C(sp³)−H bonds originating from light-weight alkanes into alkyl sulfinates, employing SO₂. This methodology grants access to a wide range of valuable organosulfur compounds bearing small aliphatic fragments. Although affordable and atom-economical, gaseous reagents are frequently disregarded due to a combination of chemical and practical considerations. In contrast, our study showcases their facile applicability as versatile reagents within the realm of organic synthesis, facilitated by the use of flow technology.

## Methods

General procedure describing the telescoped synthesis of alkyl sulfones: To a nitrogen-purged, screw-capped vial, fitted with a rubber septum and charged with NaDT (14.65 mg, 6 μmol, 3 mol%) degassed CH₃CN is added (2.4 mL), followed by aqueous SO₂ (6 wt%, 0.6 mL, 0.6 mmol, 3 equiv.). The SO₂ solution is charged in a gastight syringe, positioned in a syringe pump and combined with a stream of propane gas (73.5 mL, 3 mmol, 15 equiv.) through a T-mixer into a filling loop, with a liquid flow rate of 0.16 mL·min⁻¹ and a propane gas flow rate of 4 mL·min⁻¹. A BPR of 2.8 bar is used during the loop filling. Additionally, to a nitrogen-purged, screw-capped vial, fitted with a rubber septum and charged with benzyl bromide (34.2 mg, 23.8 μL, 0.2 mmol, 1 equiv.) and 2,6-lutidine (214 mg, 232.0 μL, 1.0 mmol, 10 equiv.) degassed CH₃CN is added (2 mL). The benzyl bromide solution was charged into a second filling loop, with one end connected to an HPLC pump and the other end connected to the outlet of the photoreactor through a T-mixer, with a shut-off valve positioned immediately before the T-mixer. Next, the SO₂ filling loop is connected to the photoreactor, the system is pressurized to 34 bar using an HPLC pump (while the shut-off valve is closed) and the reaction mixture is pumped over the Signify Eagle reactor (365 nm, 144 W output power, FEP capillary: 0.5 mm ID, 2.8 mL) at a flow rate of 0.046 mL·min⁻¹, resulting in a residence time of 1 h. At the same time, the benzyl bromide filling loop is also pressurized to 34 bar using an HPLC pump. When the reaction mixture reaches the T-mixer, the shut-off valve is open and the benzyl bromide solution is pushed at a flow rate of 0.031 mL·min⁻¹, resulting in 3:1 SO₂:benzyl bromide ratio and a total flow rate of 0.077 mL·min⁻¹. This mixture is pumped into a final reactor loop (4.6 mL), resulting in a residence time of 1 h. Then, the collected outflow is transferred to a separatory funnel, diluted with water (20 mL) and extracted with DCM (3 × 20 mL). The combined organic layers are dried over MgSO₄ and evaporated in vacuo. The crude mixture was purified by flash column chromatography (100% n-pentane to n-pentane 80:20 AcOEt) to afford

product 1a and 1b (80:20 ratio determined by [1]H NMR analysis of the crude reaction mixture), (34.1 mg, 86%) as a clear oil.

## Data availability

The data supporting the results of the article, including optimization studies, experimental procedures, compound characterization and scale-up procedures are provided within the paper and its Supplementary Information. Additional data are available from the corresponding author upon request.

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

## Acknowledgements

We are grateful to have received generous funding from the European Union H2020 under the HORIZON-HLTH-2021-IND-07 (SusPharma,

101057430, T.N., D.N., N.K.), the ERC CoG (FlowHAT, No. 101044355, T.N.), and the FETopen (FLIX, No. 862179, T.N. and F.R.).

## Author contributions

D.N., F.R., and N.K. designed the project, with input from T.N. D.N., and F.R. performed and analyzed the synthetic experiments with input from N.K. and T.N. All authors provided input during the progress meetings. D.N., F.R., and T.N. wrote the manuscript with input from all the authors.

## Competing interests

The authors declare no competing interest.
