## [Peer Review File · Nature Communications]

C(sp³)–H sulfinylation of light hydrocarbons with sulfur dioxide via hydrogen atom transfer photocatalysis in flowREVIEWER COMMENTS

Reviewer #1 (Remarks to the Author):

This manuscript reported a photocatalytic C(sp³)-H sulfonylation of lightweight alkanes with sulfur dioxide using hydrogen atom transfer photocatalysis in flow. The mild reaction conditions of this reaction deliver the diverse sulfur-containing organic compounds via the corresponding sulfinates.

However, the main content of this work somehow is an extension of the well explored flow chemistry in hydrogen atom transfer photocatalysis by the Noel group and many other groups. Moreover, the conversion of sulfur dioxide surrogates into the corresponding Sulfones, sulfonamides, and sulfonate esters has been widely reported by many groups all over the world in recent years, thus the novelty of current work is significantly diminished. Obviously, the current methodology combined the known flow technology with sulfur dioxide insertion reaction.

Overall, although the results are interesting and the manuscript is well-prepared, the impact, novelty, and significance of this work does not reach the level of that expected in a journal of this caliber. Publication in a more specialized journal is suggested.

Reviewer #2 (Remarks to the Author):

The manuscript by Noel and coworkers presents a practical and efficient approach based on hydrogen atom transfer photocatalysis to activate volatile alkanes, including isobutane, butane, propane, ethane, and methane. The resulting nucleophilic radicals undergo sulfonylation by reacting with SO₂ to yield the corresponding sulfinic acid. Under basic conditions, the obtained sulfinic acids are converted into sulfinate, serving as a platform for the preparation of various sulfur-containing organic compounds, such as sulfones, sulfonamides, and sulfonate esters. Importantly, this process can be safely and easily conducted using flow technology. The manuscript is well-written, and the results are intriguing with potential applications in medicinal chemistry and fine chemical synthesis. I recommend accepting this manuscript for publication in Nat. Comm. with minor revisions as outlined below:

- The authors mentioned "sulfinates" as the product of the process, but it appears that sulfinic acid is the initial product of the addition of the alkyl radical to SO₂ (as depicted in the schemes). I suggest the authors provide evidence, such as MS analysis, for the formation of sulfinic acid or attempt to isolate it to clarify this point.
- The functionalization of sulfinic acids via sulfinate to obtain various S-functional groups is interesting. However, it would be worthwhile to explore the reaction under basic conditions with an electrophilic fluorinating agent, such as NFSI, to produce alkylsulfonyl fluorides. This would enhance the synthetic utility of the flow procedure.
- As a minor suggestion, I recommend adjusting the style of the optimization table to enhance readability, including one entry for each condition.
- For compound 10, it would be helpful to indicate the diastereomeric ratio for the major regioisomer as a note in Scheme 1's caption.

Reviewer #3 (Remarks to the Author):

C(sp³)-H Sulfinylation of light alkanes with sulfur dioxide to construct sulfur-containing scaffolds remains significant challenges due to the practical difficulties associated with gas-liquid reaction system, as well as the inert nature, reaction selectivity, solvent compatibility, and solubility of light hydrocarbons. Traditionally, the reactions often demand harsh conditions such as high temperatures and pressures. In this study, Professor Noël and co-workers present a practical and efficient approach for the sulfinylation of light hydrocarbons with SO₂ using flow technology and decatungstate HAT photocatalysis. Gaseous alkanes such as isobutane, butane, propane, ethane, and methane can be efficiently activated by HAT photocatalysis to selectively produce nucleophilic radicals that which react with SO₂ to yield sulfinic acids. The intermediates can be easily trapped by alkyl bromides to afford diverse sulfur-containing organic compounds including valuable sulfones, sulfonamides, and sulfonate esters. The synthetic utility of this approach is further demonstrated through the late-stage functionalization of medicinally relevant organic scaffolds. This work provides an unprecedented, practical and promising method for the selective C(sp³)-H sulfinylation of light alkanes while addressing the challenges in this field. The literatures are comprehensively covered and discussed and the manuscript is well-organized. The analysis, interpretation, and discussion are reasonable and support the conclusions and claims made in this study. I believe that this work represents a significant innovation in the field and is of great importance. Therefore, I recommend its publication in *Nature Communications* with minor revisions addressing the following questions:

1. In addition to α -unsaturated group substituted alkyl bromides (Scheme 1), can α -saturated alkyl substituted ones also be suitable for the reactions?
2. The corresponding position of footnote [a] should be marked in Table 1.
3. The compound numbers **2-6** mentioned in line 126 have not been indicated in Scheme 1.

REVIEWER COMMENTS

Reviewer #1 (Remarks to the Author):

This manuscript reported a photocatalytic C(sp³)-H sulfonylation of lightweight alkanes with sulfur dioxide using hydrogen atom transfer photocatalysis in flow. The mild reaction conditions of this reaction deliver the diverse sulfur-containing organic compounds via the corresponding sulfinates. However, the main content of this work somehow is an extension of the well explored flow chemistry in hydrogen atom transfer photocatalysis by the Noel group and many other groups. Moreover, the conversion of sulfur dioxide surrogates into the corresponding Sulfones, sulfonamides, and sulfonate esters has been widely reported by many groups all over the world in recent years, thus the novelty of current work is significantly diminished. Obviously, the current methodology combined the known flow technology with sulfur dioxide insertion reaction. Overall, although the results are interesting and the manuscript is well-prepared, the impact, novelty, and significance of this work does not reach the level of that expected in a journal of this caliber. Publication in a more specialized journal is suggested.

We respectfully disagree with the assertion that our work lacks significance. The utilization of volatile alkanes in synthetic organic chemistry represents a formidable challenge, as acknowledged by researchers such as Prof. Robert Bergman (Nature 2007, 138, 391-392) and Prof. John Hartwig (JACS 2016, 138, 2-24):

- “Remaining unsolved, but increasingly important due to the production of shale gas, is the original goal: the mild and selective conversion of methane and light hydrocarbons to functionalized feedstocks” (statement Hartwig, JACS 2016)*
- “[Natural gas] is a vast, low-cost feedstock of hydrocarbons that remains untapped as a raw material, simply because there has been no easy way to turning it into synthetically useful compounds” (Statement Bergman, Nature 2007)*

Our research makes a meaningful contribution to this complex field by addressing significant gaps in knowledge. Specifically, the broad scope, high yields, and scalability of our flow platform demonstrate our ability to effectively incorporate small alkyl fragments into valuable organic compounds. We firmly believe that our findings hold substantial importance and warrant further scrutiny and exploration, a topic we are currently working on.

Reviewer #2 (Remarks to the Author):

The manuscript by Noel and coworkers presents a practical and efficient approach based on hydrogen atom transfer photocatalysis to activate volatile alkanes, including isobutane, butane, propane, ethane, and methane. The resulting nucleophilic radicals undergo sulfonylation by reacting with SO₂ to yield the corresponding sulfinic acid. Under basic conditions, the obtained sulfinic acids are converted into sulfinate, serving as a platform for the preparation of various sulfur-containing organic compounds, such as sulfones, sulfonamides, and sulfonate esters. Importantly, this process can be safely and easily conducted using flow technology. The manuscript is well-written, and the results are intriguing with potential applications in medicinal chemistry and fine chemical synthesis. I recommend accepting this manuscript for publication in Nat. Comm. with minor revisions as outlined below:

- The authors mentioned "sulfonates" as the product of the process, but it appears that sulfinic acid is the initial product of the addition of the alkyl radical to SO₂ (as depicted in the schemes). I suggest the authors provide evidence, such as MS analysis, for the formation of sulfinic acid or attempt to isolate it to clarify this point.

We thank the reviewer for this instructive comment. The reaction between propane and SO₂, in the presence of TBADT, was run without subjecting the crude to the second step (base and alkyl bromide). The outflow of the reactor was collected and analysed by ¹H and ¹³C NMR, revealing a mixture of sulfinic acid and sodium sulfinate salt.^{1,2} When this crude was further acidified with 1M HCl and extracted into the aqueous layer, the ¹H and ¹³C NMR spectra showed peaks corresponding to isopropane sulfinic acid and n-propane sulfinic acid, in accordance with literature.¹ This suggests that sulfinic acid is indeed the initial product of the addition of the alkyl radical to SO₂, along with the corresponding sulfinate salt. However, after deprotonation by the base, the sulfinate is the species that performs the following S_N2 reaction. All the analysis could be found in Chapter 6 of the SI, Identification of reaction intermediates.

- The functionalization of sulfinic acids via sulfinate to obtain various S-functional groups is interesting. However, it would be worthwhile to explore the reaction under basic conditions with an electrophilic fluorinating agent, such as NFSI, to produce alkylsulfonyl fluorides. This would enhance the synthetic utility of the flow procedure.

*Due to the volatility and water sensitivity of sulfonyl fluorides derived from gaseous alkanes, the isolation and quantification of these products proved to be challenging. Instead, the sulfonyl fluorides obtained from the reaction of ethane and SO₂, with the follow up addition of SelectFluor, were immediately engaged in a reaction to form a sulfonic ester. For this purpose, the substrate used to generate compound **34** was chosen, to highlight the usefulness of forming intermediate sulfonyl fluorides en route to synthetically complex products, and to provide a comparison between using sulfonyl chlorides and sulfonyl fluorides to access the same molecule. Crude ¹H and ¹⁹F NMR of the intermediate sulfonyl fluoride are present at the end of the SI. Compound **34** was obtained in a 20% yield using ethane sulfonyl fluoride as an intermediate.*

- As a minor suggestion, I recommend adjusting the style of the optimization table to enhance readability, including one entry for each condition.

We appreciate the suggestion and have indeed attempted its implementation. However, upon review, we observed that the resultant table became considerably lengthy, potentially compromising its

readability. Consequently, after careful consideration, we have opted to retain the original version, as it appeared to offer the most optimal balance for presentation.

- For compound 10, it would be helpful to indicate the diastereomeric ratio for the major regioisomer as a note in Scheme 1's caption.

We thank the reviewer for the comment, the caption was adjusted accordingly.

Reviewer #3 (Remarks to the Author):

C(sp³)-H Sulfonylation of light alkanes with sulfur dioxide to construct sulfur-containing scaffolds remains significant challenges due to the practical difficulties associated with gas-liquid reaction system, as well as the inert nature, reaction selectivity, solvent compatibility, and solubility of light hydrocarbons. Traditionally, the reactions often demand harsh conditions such as high temperatures and pressures. In this study, Professor Noël and co-workers present a practical and efficient approach for the sulfonylation of light hydrocarbons with SO₂ using flow technology and decatungstate HAT photocatalysis. Gaseous alkanes such as isobutane, butane, propane, ethane, and methane can be efficiently activated by HAT photocatalysis to selectively produce nucleophilic radicals that which react with SO₂ to yield sulfinic acids. The intermediates can be easily trapped by alkyl bromides to afford diverse sulfur-containing organic compounds including valuable sulfones, sulfonamides, and sulfonate esters. The synthetic utility of this approach is further demonstrated through the late-stage functionalization of medicinally relevant organic scaffolds. This work provides an unprecedented, practical and promising method for the selective C(sp³)-H sulfonylation of light alkanes while addressing the challenges in this field. The literatures are comprehensively covered and discussed and the manuscript is well-organized. The analysis, interpretation, and discussion are reasonable and support the conclusions and claims made in this study. I believe that this work represents a significant innovation in the field and is of great importance. Therefore, I recommend its publication in *Nature Communications* with minor revisions addressing the following questions:

1. In addition to α -unsaturated group substituted alkyl bromides (Scheme 1), can α -saturated alkyl substituted ones also be suitable for the reactions?

We thank the reviewer for this valuable comment. Indeed, we used activated electrophiles for the substitution reaction with sulfinates. Under our standard reaction conditions, the reaction between ethane and SO₂ was run with the subsequent addition of methyl iodide in the second step, and the desired compound was obtained in 30% yield. We also mention the use of alkyl iodide in the manuscript (line 121).

2. The corresponding position of footnote [a] should be marked in Table 1.

We thank the reviewer for carefully checking our manuscript. We adjusted the manuscript accordingly.

3. The compound numbers **2-6** mentioned in line 126 have not been indicated in Scheme 1.

We adjusted the sentence to refer to the products of the reaction.

References:

1. Gu, D.; Harpp, D. N. The reaction of mercaptans with dimethyldioxirane. A facile synthesis of alkanesulfinic acids. *Tetrahedron Letters* **1993**, *34* (1), 67-70.
2. Meyer, A. U.; Straková, K.; Slanina, T.; König, B. Eosin Y (EY) Photoredox-Catalyzed Sulfonylation of Alkenes: Scope and Mechanism. *Chemistry – A European Journal* **2016**, *22* (25), 8694-8699.

REVIEWERS' COMMENTS

Reviewer #2 (Remarks to the Author):

The authors have diligently addressed the concerns raised by this reviewer. Based on the thorough revisions, I highly recommend accepting this manuscript for publication in Nature Communications.

Reviewer #3 (Remarks to the Author):

The authors have responded to all the reviewer's concerns and revised the manuscript accordingly. I recommend that the revised manuscript be published in Nature communications.